# Single-cell RNA sequencing reveals cellular and molecular heterogeneity in fibrocartilaginous enthesis formation

**Tao Zhang**[1,2,3†], **Liyang Wan**[1,2,3†], **Han Xiao**[1,2,3], **Linfeng Wang**[1,2,3], **Jianzhong Hu**[2,3,4], **Hongbin Lu**[1,2,3]*

[1]Department of Sports Medicine, Xiangya Hospital Central South University, Changsha, China; [2]Key Laboratory of Organ Injury, Aging and Regenerative Medicine of Hunan Province, Changsha, China; [3]National Clinical Research Center for Geriatric Disorders, Xiangya Hospital, Central South University, Changsha, China; [4]Department of Spine Surgery and Orthopaedics, Xiangya Hospital, Central South University, Changsha, China

**\*For correspondence:**
hongbinlu@hotmail.com

[†]These authors contributed equally to this work

**Competing interest:** The authors declare that no competing interests exist.

**Abstract** The attachment site of the rotator cuff (RC) is a classic fibrocartilaginous enthesis, which is the junction between bone and tendon with typical characteristics of a fibrocartilage transition zone. Enthesis development has historically been studied with lineage tracing of individual genes selected a priori, which does not allow for the determination of single-cell landscapes yielding mature cell types and tissues. Here, in together with open-source GSE182997 datasets (three samples) provided by Fang et al., we applied Single-cell RNA sequencing (scRNA-seq) to delineate the comprehensive postnatal RC enthesis growth and the temporal atlas from as early as post-natal day 1 up to postnatal week 8. And, we furtherly performed single-cell spatial transcriptomic sequencing on postnatal day 1 mouse enthesis, in order to deconvolute bone-tendon junction (BTJ) chondrocytes onto spatial spots. In summary, we deciphered the cellular heterogeneity and the molecular dynamics during fibrocartilage differentiation. Combined with current spatial transcriptomic data, our results provide a transcriptional resource that will support future investigations of enthesis development at the mechanistic level and may shed light on the strategies for enhanced RC healing outcomes.

## Editor's evaluation

This paper represents a valuable single-cell level analysis of tendon enthesis development. The study allows further understanding of this specific process with clinical implications. The authors provided convincing evidence for the heterogeneity of postnatal enthesis growth and the molecular dynamics and signaling networks during enthesis formation.

## Introduction

RC and its enthesis are essential components of the shoulder, which are critical in facilitating coordinated shoulder movements and stability (*Schett et al., 2017*). Compared to other RC tissues, the attachment site of supraspinatus (SS) tendon is vulnerable to injury and difficult to achieve complete regeneration, due to its high heterogeneity in composition and structure (*Nourissat et al., 2015*; *Schett et al., 2017*). Histologically, the attachment site of the SS tendon is a classic fibrocartilaginous enthesis, also termed BTJ (*Chen et al., 2021a*; *Rossetti et al., 2017*). In its native state, the fibrocartilaginous enthesis exhibits gradations in tissue organization, cell phenotype, and matrix composition

(*Moffat et al., 2008*; *Rossetti et al., 2017*). Fibrocartilaginous entheses manage to disperse stress and facilitate load transfer between vastly different materials like tendons and bone, with modulus ranging from 200 MPa to 20 GPa (*Rossetti et al., 2017*). Unfortunately, the intrinsic regenerative capacity of fibrocartilaginous enthesis is not well-understood, which limits the exploitation of the best and most rigorously proven early intervention programs (*Derwin et al., 2018*; *Schett et al., 2017*; *Xiao et al., 2022*). Therefore, understanding the complex process of the enthesis morphogenesis and maturation during development may inform strategies for enhanced BTJ healing.

Currently, the mechanism underlying the growth of the enthesis fibrocartilage is less understood. Details are scarce, but the fibrocartilage layer is formed by a pool of site-specific progenitor cells, and initially organizes as an unmineralized cartilaginous attachment unit (*Jensen et al., 2018*). Such development pattern shares an overlapping biological behavior with the growth plate, which is a process of mesenchymal stem cells differentiating into chondrogenic cells and then sequentially into fibrocartilage cells (*Killian, 2022*). A unique enthesis progenitor pool has been identified since the embryonic stages, as cells sandwiched between primary cartilage and tendon, expressing a mixed transcriptome of both chondrogenic and tenogenic genomic features (Scleraxis and SRY-related transcription factor) (*Blitz et al., 2013*; *Sugimoto et al., 2013*). These enthesis progenitors can differentiate into either chondrocytes or tendon fibroblasts, under the regulation of Krüppel-like (*Klf*) transcription factors (*Kult et al., 2021*). In the later embryonic and postnatal stages, cells from the enthesis progenitor pool ultimately either differentiate into or are replaced by a *Hh*-positive cell population marked by *Gli1* and *Ptch1* (*Felsenthal et al., 2018*; *Schwartz et al., 2015*). *Gli1*[+] cells and their progenies are retained in the enthesis region throughout postnatal development and eventually populate the entire fibrocartilage region between tendon and bone, thereby contributing to enthesis growth (*Felsenthal et al., 2018*; *Jensen et al., 2018*; *Schwartz et al., 2015*). However, compared with our in-depth understanding of enthesis development during the embryonic stage, the cell-type composition and distribution in the enthesis at different postnatal stages, as well as biochemical markers for enthesis stem cell progenitor used in tissue engineering, remain to be not well-understood and require novel methods for elucidation.

As is proved, the normal enthesis maintains a gradient of cell phenotypes, from tendon fibroblast to chondrocyte then to mineralizing chondrocyte and to osteoblast/osteocyte (*Chen et al., 2021b*; *Moffat et al., 2008*; *Rossetti et al., 2017*). It is unclear how this gradient in cell phenotypes develops and how it is regulated by the local environment (e.g. extracellular matrix, muscle loading, and growth factors) (*Derwin et al., 2018*). scRNA-seq is a powerful method to analyze various cell types and provide insights into tendon enthesis postnatal development (*Gulati et al., 2020*; *Kult et al., 2021*). Recently, Fang Fei et al., reported an exciting single-cell work to define the enthesis cell transcriptomes at postnatal day 11, 18, and 56, revealed the clonogenicity and multipotency of enthesis *Gli1*+ progenitors (*Fang et al., 2022*). However, compared to the abundant and growing amount of single-cell resources in bone, cartilage, and tendon, more enthesis single-cell resources are needed to cover long-range timepoints of BTJ development or entheseal diseases. Here, we applied single-cell transcriptomics along with the spatial transcriptomic sequencing to analyze the cellular and molecular dynamics during postnatal tendon enthesis growth. The results provided here for deciphering postnatal tendon enthesis development may facilitate future studies of enthesis regeneration.

## Results

### The development of fibrocartilage in the enthesis occurs postnatally

The heterogeneity of the fibrochondrocytes in enthesis development has been an open question. Aiming to evaluate the postnatal development pattern of enthesis fibrocartilage, we first stained the shoulder sections and compared the morphological parameters of the enthesis cells at P1, P3, P7, P14, P28, and P56 (*Figure 1a*). At postnatal day 1, an articular cavity was formed and the supraspinatus tendon (ST) was observed attached to the humeral head. The cells at the ST attachment site were highly dense and homogeneous, and were visibly different from tendon cells and the primary cartilage cells inside the humeral head, suggesting that postnatal enthesis is formed by site-specific progenitor cells since the embryonic stage. At P1, P3, and P7, the fibrocartilage layer and subchondral bone could hardly be visualized. Fibrocartilage was not evident at the enthesis of the mouse rotator cuff until 2–3 weeks after birth. To examine the cellular morphological changes, we further measured

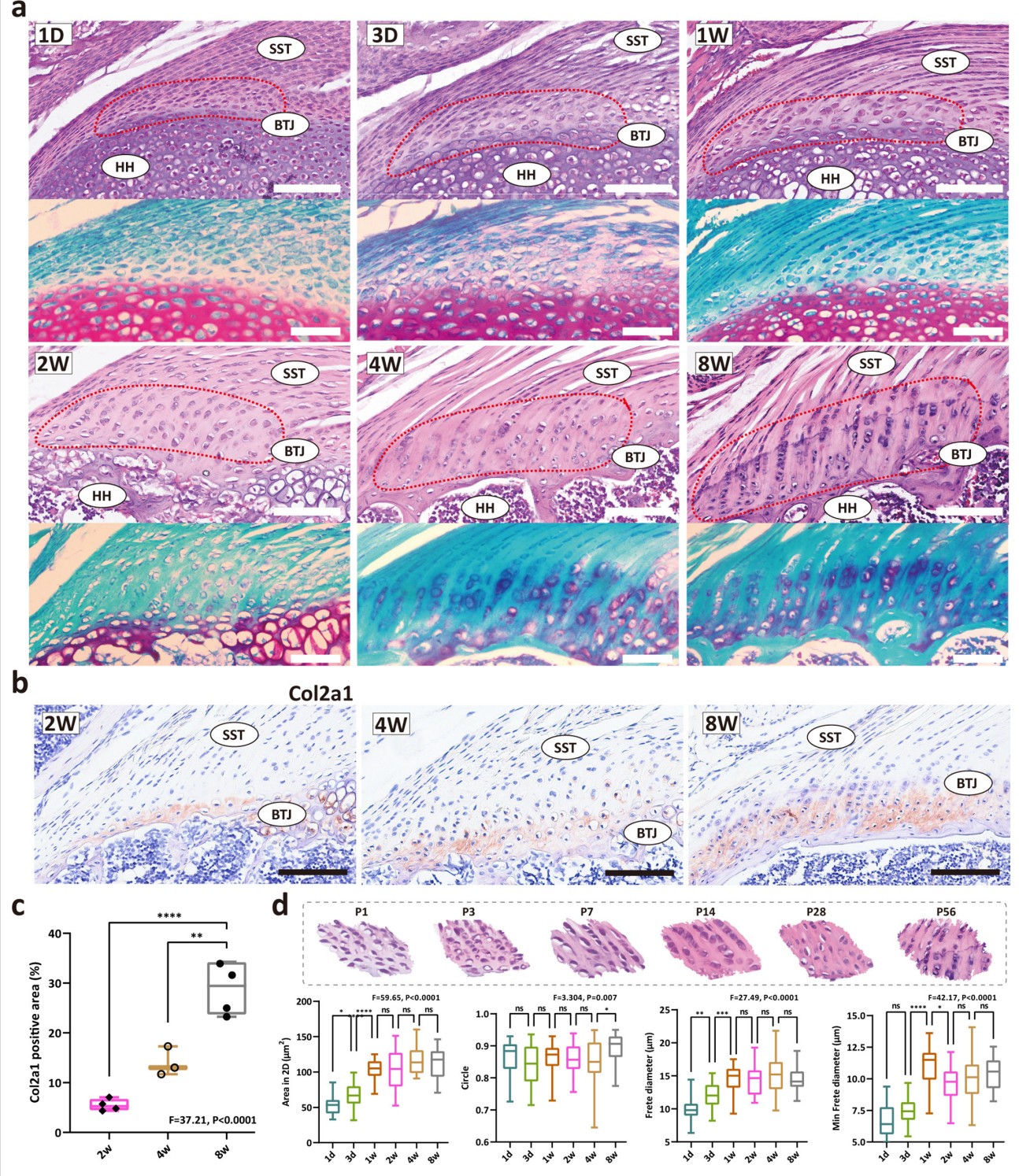

**Figure 1.** The development of fibrocartilage in the enthesis occurs postnatally. (**a**) H&E staining of P1, P3, P7, P14, P28, and P58 mouse supraspinatus tendon entheses (n=4). Scale bars, 100 μm. (**b**) Immunohistochemistry images of cartilage-abundant collagen II at P14, P28 and P56. Scale bars, 100 μm. (**c**) Comparison of the col2a1 positive area derived from IHC results. Error bars represent SEM. N=3-4. **p<0.01, ****p<0.0001. (**d**) Comparison of the morphological parameters (2D area, roundness, frete diameter, and minimal frete diameter) between P1, P3, P7, P14, P28, and P56. Error bars represent SEM. *p<0.05, ***p<0.001, ****p < 0.0001.

the 2D parameters (including 2D area, roundness, frete diameter, and minimal frete diameter) of enthesis cells, statistics results demonstrated that cell size remarkably increased during postnatal development, from postnatal day 7–14 (*Figure 1d*). After postnatal day 14, the enthesis cells were observed as typical chondrocyte phenotype, column-like stacked alongside the direction of tendon fiber, with more prominent patterns at P28 and P56. From toluidine blue/fast green stainings, fibro-chondrocytes can be observed since P14, with a larger cellular size in comparison with condensed primary chondrocytes in P7. The extracellular matrix (ECM) of fibrocartilage with mineralization can be observed after P28. Type 2 collagen is a major ECM component of cartilage, thus we stained enthesis with col2a1 antibody. The IHC results show the col2a1 protein levels are relatively low at P14, and significantly increase after P28 (*Figure 1b and c*).

## Unbiased clustering identified known cell populations in postnatal enthesis development

To determine the cellular composition of the developing enthesis, we integrated our dataset with open-source GSE182997 datasets (three samples) provided by Fang et al., (*Figure 2a*). After the elimination of doublets, dead, and apoptotic cells, blood cells (erythrocytes and progenitors), endothelial cells, immune cells (B cells and T cells), myeloid cells, and growth plate chondrocytes, we got high-quality transcriptomic data from 8368 single cells, including 1285 P1 cells, 4059 P7 cells, 918 P11 cells, 307 P14 cells, 329 P18 cells, 568 P28 cells, and 897 P28 cells (*Figure 2—figure supplement 1*). Unbiased clustering based on Uniform Manifold Approximation and Projection (UMAP) identified major cell populations. Based on the differentially expressed genes (DEGs) (*Figure 2—figure supplement 2*), all the cell clusters were annotated, including BTJ chondrocytes, BTJ tendons, Tenocytes, Osteocytes, Enthesoblasts, Tenoblasts, and Mesenchymal progenitors (*Figure 2b*). We used an entropy-based metric (ROUGE) for assessing the purity of single-cell clusters, all these cell subtypes achieved high ROGUE values of >0.9 (*Liu et al., 2020*), which suggested accuracy in unsupervised cell clustering (*Figure 2—figure supplement 2*). Generally, enthesis progenitors had relatively high expression of *Ly6a*, *Cd44*, and *Pdgfrα*, in agreement with progenitors found in tendons and bone marrow (*Harvey et al., 2019*; *Tikhonova et al., 2019*). Enthesoblasts were defined based on their relatively decreased stemness transcriptional signatures, as well as co-expressed tenogenic (e.g. Scx, Col1a1) and chondrogenic markers (e.g. Sox9, Acan) (*Jensen et al., 2018*). The correlation analysis showed the consistency between our datasets and Fang et al., reported datasets (*Figure 2—figure supplement 2*). The barplot diagram showed cell identity change after birth (*Figure 2c*).

We used the spatial transcriptome data of mice 1 day after birth to verify the cell population defined in our P1 single-cell dataset, and the enthesis chondrocytes group was consistent with the spatial anatomical position of the bone-tendon junction (*Figure 2e*). We checked the previously reported enthesis marker genes *Sox9* and *Scx*, as well as enthesis-specific ECM genes (*Col2a1*), which were ubiquitously expressed in BTJ chondrocyte cells (*Figure 2e*). We then performed an immunofluorescence assay to validate the spatial distribution of enthesis-related genes, we found that *Sox9*[+] and *Scx*[+] cells were detected in the enthesis area, mostly in the neonatal stage and significantly decreased in postnatal weeks 2 and 4, as expected (*Figure 2f*).

## Identifying developmental trajectories for tenocytes, chondrocytes, and osteocytes differentiation in enthesis

We next sought to investigate the trajectory and regulatory genes to govern postnatal bone-tendon junction cell development. We first predicted the differentiation state of each cell group from scRNA-seq data by using Cytotrace (*Gulati et al., 2020*). The Cytotrace results showed that across all cell clusters, the 'stemness' degree of mesenchymal progenitors is higher than other cell types (*Figure 3a*). Among tendon and enthesis-associated cell groups, CytoTRACE scores of the enthesis cells (BTJ chondrocytes cells) skewed toward a moderate predicted stem potential, which was slightly higher than that in tendon cells, suggestive of a higher degree of stemness for enthesis cells developing into fibrocartilage cells (*Figure 3b*).

We next implemented RNA velocity analysis. The trajectory results showed enthesoblasts originated from mesenchymal progenitors, and enthesoblasts had a greater chance to branch into enthesis chondrocytes (*Figure 3c*). Next, we employed developmental potential calculated by Cytotrace to compute the terminal states among all the cell groups by Cellrank, which is a toolkit based on Markov

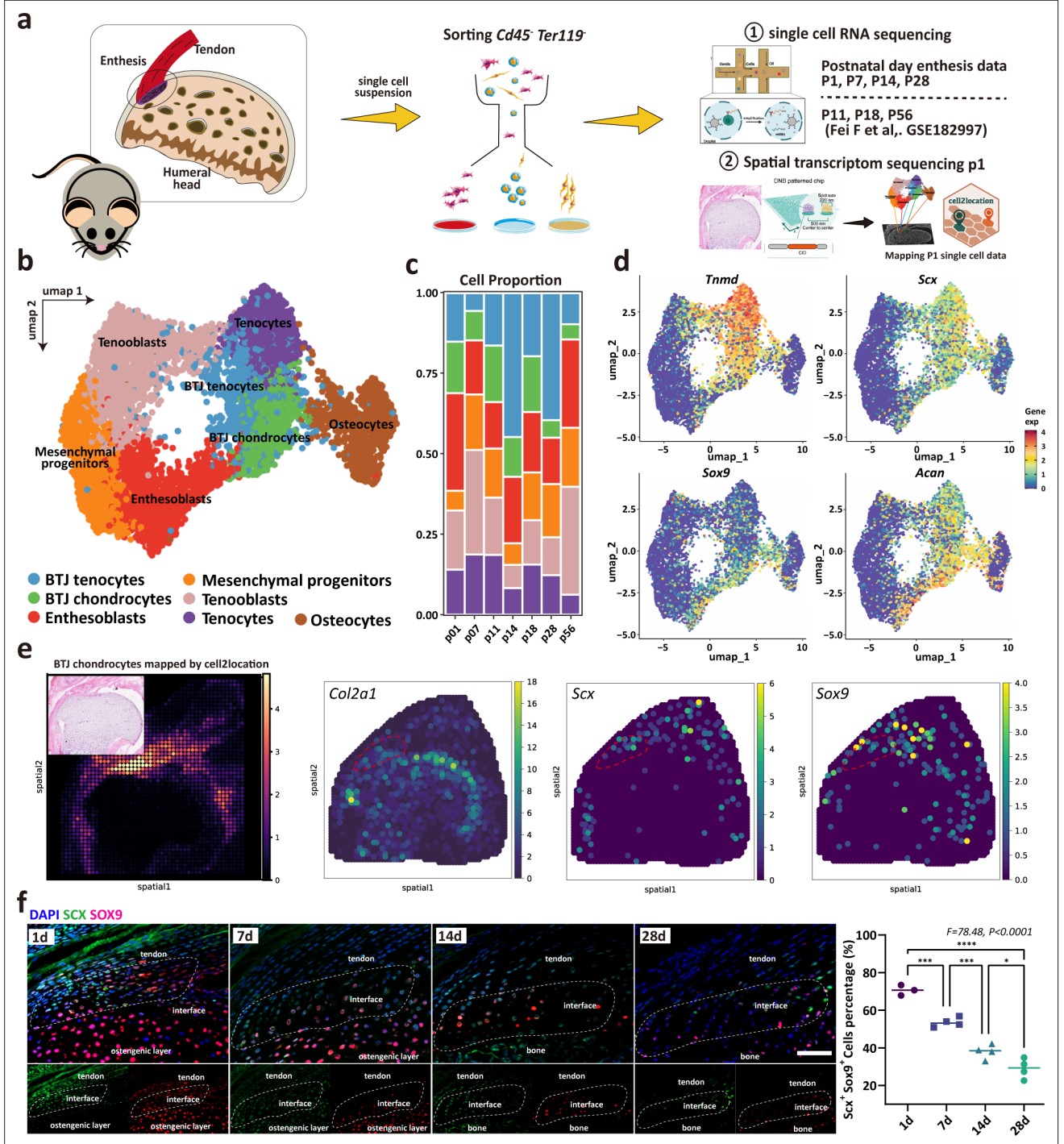

**Figure 2.** Unbiased clustering identified Known Cell Populations in postnatal enthesis development. (**a**) Schematic workflow of the study design. (**b**) Distributions of seven cell clusters on UMAP plot, including bone-tendon junction (BTJ) chondrocytes, BTJ tendons, tenocytes, osteocytes, enthesoblasts, tenoblasts, and mesenchymal progenitors. (**c**) Fractions of cell clusters in enthesis development at P1, P7, P11, P14, P18, P28, and P56. (**d**) The average expression of curated feature genes for previously reported enthesis marker genes and enthesis-specific extracellular matrix (ECM) genes. (**e**) Spatial transcriptomic spot map revealing the expression of chondrocyte marker genes in each spatial spot. (**f**) Representative immunofluorescence staining to validate the spatial distribution of *Sox9*+ and *Scx*+ cells in the enthesis area, at P1, P7, and P14. Scale bars, 100 μm. N=3-5. *p<0.05, ***p<0.001, ****p<0.0001.

The online version of this article includes the following figure supplement(s) for figure 2:

**Figure supplement 1.** Technical and quality control measures for each single-cell RNA sequencing (scRNA-seq) datasets and single-cell transcriptomic dataset, related to Figure 2.

*Figure 2 continued*

**Figure supplement 2.** Top differentially expressed genes per cluster in the dataset, related to *Figure 2e*.

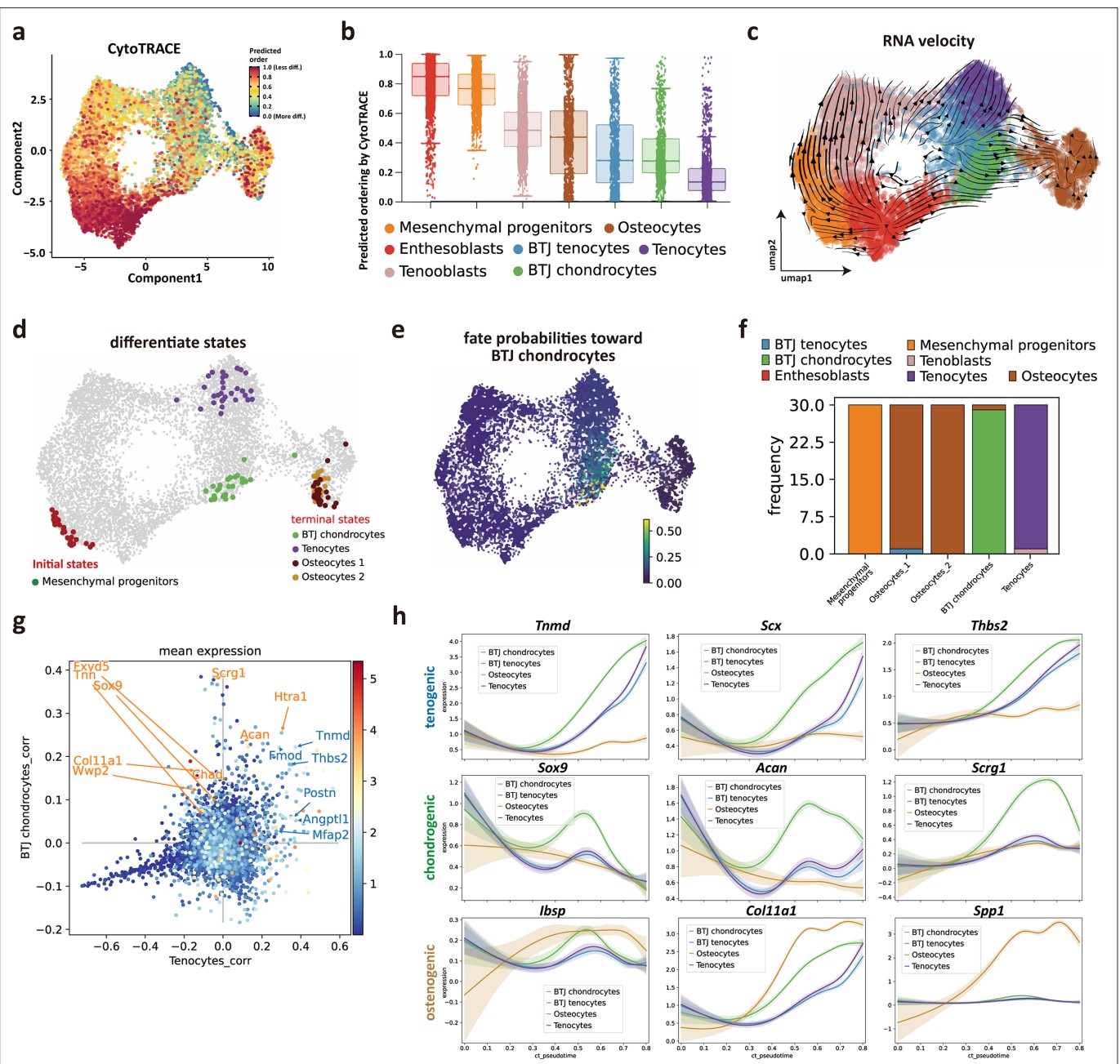

**Figure 3.** Identifying developmental trajectories for tenocytes, chondrocytes, and osteocytes differentiation in enthesis. (**a**) Uniform Manifold Approximation and Projection (UMAP) plot of enthesis single-cell RNA sequencing (scRNA-seq) data overlaid with CytoTRACE scores. (**b**) Boxplot of predicted differentiation score distributions for each cell cluster. (**c**) Results of RNA velocity analysis show that postnatal enthesis fibrocartilage origin from mesenchymal progenitors, instead of tendon cells. (**d**) Cellrank identified three differentiation terminal cell types, including bone-tendon junction (BTJ) chondrocytes, tenocytes, and osteocytes. (**e**) Fate probabilities uncovered putative BTJ chondrocytes lineage drivers. (**f**) Distribution over cluster membership for each of the cells assigned to a certain terminal state. (**g**) Genes that correlate positively with the BTJ chondrocyte fate correlate moderately with the tendon fate and vice versa. (**h**) Representative tenogenic, chondrogenic, and osteogenic gene expression dynamics along four terminal differentiation trajectories.

state modeling (*Lange et al., 2022*), and four clusters of cells were predicted as the terminal differentiate clusters: BTJ chondrocytes, tenocytes, and two subsets of osteocytes (*Figure 3d*), suggesting that the differentiation profile of these cell groups was separative from each other. The fate probabilities analysis showed that only BTJ chondrocytes contributed to the fate of enthesis chondrocytes (*Figure 3e and f*), which was consistent with RNA velocity, suggesting that the fibrocartilage in postnatal enthesis origin from enthesis site-specific progenitors, instead of tendon cells. However, we found BTJ chondrocytes correlate slightly with tenocytes, because the genes that correlate positively with the BTJ chondrocyte fate correlate moderately with the tendon fate and vice versa (*Figure 3g*).

To explore gene expression dynamics along the trajectories, we measured the dynamics of genes in pseudotime along the differentiation trajectories of these three differentiation terminal cell types. We found that the expressions of known differential regulator genes were upregulated significantly higher in enthesis-associated trajectories, such as *Scx*, *Sox9*, and *Tnmd*, which were confirmed indispensable for enthesis formation. We also found genes related to cartilage ECM (*Acan*, *Scrg1*) were upregulated along the pseudotime, and highly expressed until the terminal differentiation state, suggesting the collagen and matrix protein synthesis were predominant in postnatal enthesis growth. Meanwhile, mineralization-related genes (*Ibsp*, *Col11a1*) were observed to increase in enthesis chondrocyte differentiation, but less than their expression levels in osteocytes (*Figure 3h*).

## Reconstruction of the trajectory and gene dynamics in BTJ chondrocytes differentiation

As enthesis progenitors differentiating into fibrochondrocytes is the pivotal step of enthesis development, we sought to determine their gene dynamics. Mesenchymal progenitors, enthesoblasts, and enthesis chondrocytes were subsets to receive trajectory analysis. We used monocle3 to identify the unsupervised pesudotime order within the three clusters of single cells (*Figure 4a*). In consistence with the differentiating order predicted by Cytotrace algorithm, the mesenchymal progenitors had the highest differential potential, and the differentiate routine starting from mesenchymal progenitors to enthesoblasts and finally to enthesis chondrocytes (*Figure 4b* and *Figure 4—figure supplement 1*). To illustrate the gene ontology changes of BTJ chondrocytes differentiation between different timepoints, we next performed time-dependent Gene Ontology (GO) enrichment analysis. The biological processes alike extracellular matrix organization, collagen fibril organization, and regulation of cellular response to growth factors stimulus were significantly upregulated with enthesis developmental time increasing to postnatal day 14, 28, and 56 (*Figure 4c* and *Figure 4—figure supplement 1c*).

To explore gene expression dynamics along the BTJ chondrocytes differentiation trajectories, we next examined gene patterns that varied with BTJ chondrocytes into 25 modules using Louvain community analysis. We found genes in Module 15 and Module 22 were expressed preferentially in BTJ chondrocytes, which were annotated for cartilage development, ossification, and biomineralization (*Figure 4e*). In contrast, stemness-related genes (*Ly6a*, *Cd34*, and *Cd44*) were downregulated along the trajectories, whose gene modules annotated for mesenchyme morphogenesis, regulation of organ formation, and regulation of mesenchymal cell proliferation (*Figure 6—figure supplement 1*).

In light of the gene dynamics along the postnatal BTJ chondrocytes differentiation, we measured the gene expression in pseudotime. The heatmap and the gene feature plots show the putative genes which driving BTJ chondrocyte differentiation, as well as the top driving genes predicted by cellrank (*Figure 4g and h*). Among the most highly significant driving genes (*Klf9*, *Fxyd5*, *Klf4*, *Mfge8*, *Sox9*, *Clec3a*, *Wwp2*, *Tnn*), we then confirmed the expression of *Klf9*, *Klf4*, *Clec3a*, *Mfge8*, and *Tnn* in single-cell spatial transcriptomics, except for previously reported *Sox9* and *Wwp2* which relative to chondrogenesis (*Blitz et al., 2013*). We found the expression of *Mfge8* and *Tnn* had not been reported, and we validated the expression of *Mfge8* and *Tnn* proteins in enthesis, as the expression of Tnn increased and reached the top at postnatal week 2 (P14), and the expression of *Mfge8* increased and maintained a higher level after birth (*Figure 4j*).

## Biological processes and regulators governing enthesis chondrocytes differentiation

To determine cellular functions across varied cell subpopulations or development stages, we then compared the GO ontology across major cell clusters. As expected, BTJ chondrocytes were enriched with genes annotated for cartilage development, extracellular matrix organization, and biomineral

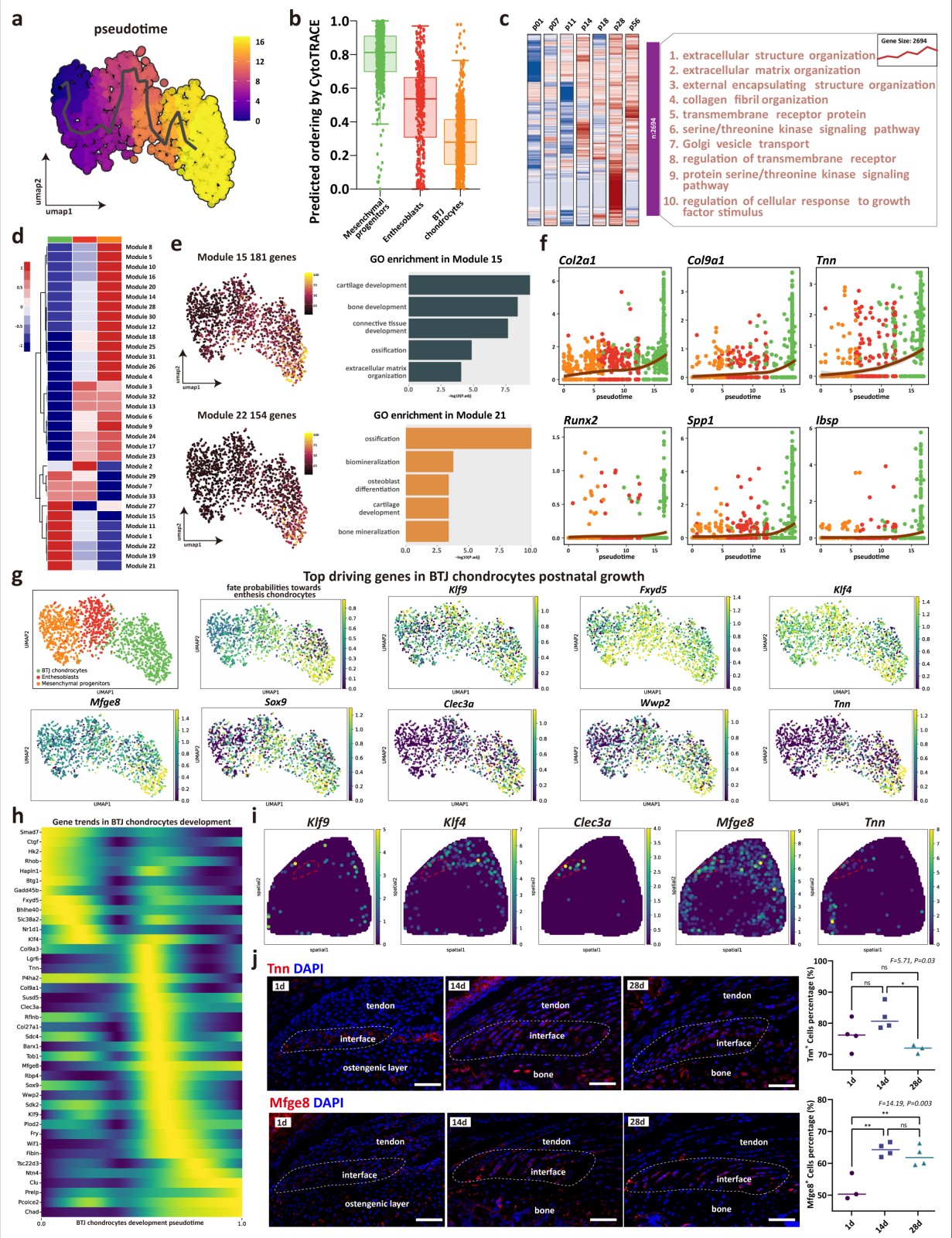

**Figure 4.** The trajectory and gene dynamics in bone-tendon junction (BTJ) chondrocytes differentiation. (**a**) Pseudotime analysis of the three clusters of mesenchymal progenitors, enthesoblasts, and enthesis chondrocytes. (**b**) Cytotrace scores and predicted ordering of the three subclusters. (**c**) Heatmap revealing the scaled expression of differentially expressed genes (DEGs) and their annotated function for each timepoint. (**d**) Heatmap showing the modules of coregulated genes grouped by Louvain community analysis. (**e**) Uniform Manifold Approximation and Projection (UMAP) plots

*Figure 4 continued on next page*

*Figure 4 continued*

and functional annotation of the representative gene modules, showing the top five gene ontology (GO) annotations of indicated biological process among each timepoint. (**f**) Line plots showing representative gene trends in modules 15 and 22, respectively. (**g**) Feature plots showing top driving genes in BTJ chondrocyte growth. (**h**) Heatmap showing the gene expression dynamics along differentiation trajectories of BTJ chondrocytes. (**i**) Spatial transcriptomic spot map reveals the expression of driving genes in each spatial spot. (**j**) Immunofluorescence shows distribution and dense level of *Tnn* and *Mfge8* during the different postnatal times. Scale bars, 100 μm. N=3-4. *p<0.05, **p<0.01.

The online version of this article includes the following figure supplement(s) for figure 4:

**Figure supplement 1.** The trajectory and gene dynamics in bone-tendon junction (BTJ) chondrocytes differentiation, related to *Figure 4e*.

tissue development, and enthesoblasts were enriched with fibroblast proliferation and developmental growth involved in morphogenesis (*Figure 5a and b*). The time-dependent analysis of each GO annotation in BTJ chondrocytes showed that biological processes alike chondrocyte proliferation, development, and biomineral tissue development decreased with time increased to P56. While the activity of collagen synthesis maintained a steady level from after birth to postnatal days 28 and 56 (*Figure 5c*).

We performed single-cell regulatory network inference and clustering (SCENIC) to investigate the gene regulatory networks that might govern enthesis growth (*Figure 5d*). The results showed that previously reported enthesis-related regulon *Sox9* was significantly expressed both in BTJ tenocytes and chondrocytes, in comparison with other cell types. We then checked the *Sox9* expression in single-cell dataset and spatial transcriptomics, the results showed that *Sox9* regulon is mostly expressed in chondrocytes and some parts of tenocytes adjunct to enthesis. Target gene analysis showed the downstream targets of *Sox9* regulon were mostly genes known associated with cartilage development (*Acan*, *Scrg1*, *Hapln1*, *Chad*) (*Figure 5e*). In-vivo validation results showed that in an early stage of postnatal growth (P1 and P7), *Sox9*-positive cells were widely located in the tendon enthesis and humeral head. And the expression of *Sox9* decreased with time increased, partly visible in the enthesis cell (*Figure 5f*). In addition, we checked the expression of *Mef2a/Mef2c*, which had been reported relative to biomineralization and chondrocyte hypertrophy (*Chen et al., 2023*; *Leupin et al., 2007*). Immunofluorescence revealed that these *Mef2a/Mef2c* positive populations were found at the SST enthesis, and were more abundant at the mid-stage of enthesis differentiation (P14-P28), consistent with the emergence of fibrochondrocytes observed in histological stainings (*Figure 5h*).

## Intercellular crosstalk signaling networks regulating enthesis postnatal growth

To seek further insights into the critical factors which may regulate the enthesis postnatal growth, we refined the CellChat and cellphoneDB input for downstream analysis (including seven clusters of BTJ chondrocytes, BTJ tendons, Tenocytes, Osteocytes, Enthesoblasts, and Mesenchymal progenitors) and performed the signaling communication analysis. Both CellChat and cellphoneDB results identified the aggregated signaling network for intercellular crosstalk. Relative active bidirectional signaling interactions among these cell subclusters revealed highly regulated cellular communications (*Figure 6a*). We then identified the signaling roles of each subcluster, the results showed that the cluster of BTJ chondrocytes predominately showed incoming patterns, as suggestive of signaling receivers (*Figure 6b*). We further identified signal components that contribute most to the incoming signaling among all these subclusters (*Figure 6c and d*).

FGF and BMP signalings are crucial to articular cartilage development, yet their specific roles in enthesis development need further investigation. To determine the important factors, we analyzed the intercellular signaling networks of FGF and BMP signalings. First, the expression pattern of FGF-FGFR signaling was noticed, as both autocrine and paracrine in BTJ chondrocyte. BTJ chondrocytes, tenocytes, and mesenchymal progenitors were leading senders of FGF signaling (*Figure 6e*). We observed *Fgfr2* expressed mostly in BTJ chondrocyte cells via *Fgf2-Fgfr2* or *Fgf9-Fgfr2*, and we validated the expression of FGFR2 protein in enthesis by immunofluorescence staining (*Figure 6f*, and *Figure 6—figure supplement 1*). The transforming growth factor-β (TGF-β) superfamily includes a family of proteins, such as TGF-βs (TGF-β1, TGF-β2, and TGF-β3) and bone morphogenetic proteins (e.g. BMP2, BMP4). In the BMP signaling network, the BTJ chondrocytes acted as critical receivers and contributors by secreting BMP ligand *Bmpr2*, especially (*Figure 6g* and *Figure 6—figure supplement 1*). In immunofluorescence results, BMPR2 was observed highly expressed in enthesis from P7 to P56. Ligand-receptor analysis points to *Bmp2* and *Bmp4* sent from mesenchymal progenitors and

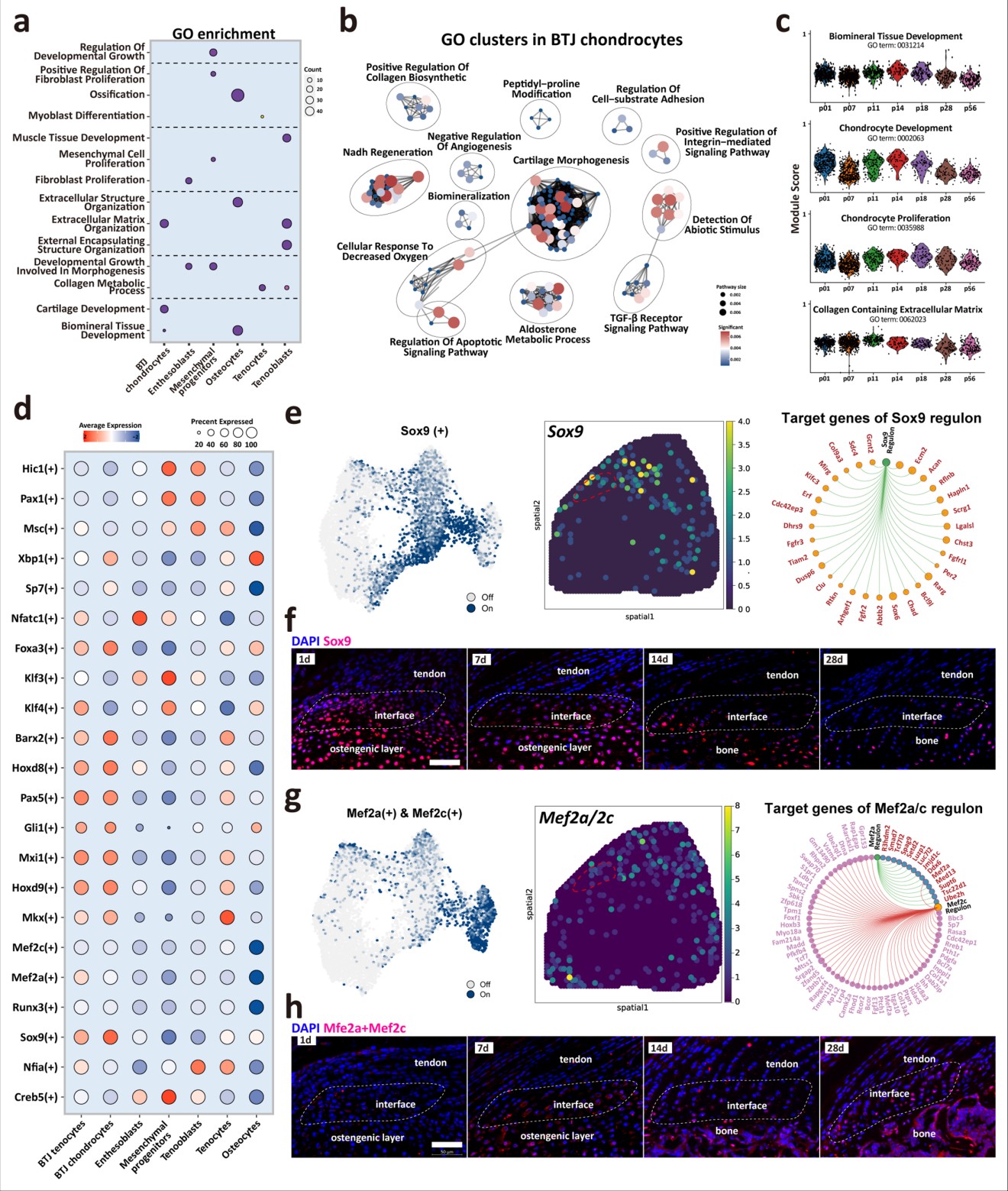

**Figure 5.** Biological processes and regulators governing enthesis chondrocytes differentiation. (**a**) Heatmap shows the typical biological processes enriched in each cellular cluster. (**b**) Dot plots show the gene ontology (GO) clusters in bone-tendon junction (BTJ) chondrocytes in enthesis development. (**c**) Time-dependent analysis of GO annotations including chondrocyte proliferation, development, and biomineral tissue development decreased with time increased. (**d**) Heatmap shows the most significant regulatory regulons in each subcluster. (**e, g**) Uniform Manifold Approximation and Projection (UMAP) plots and spatial expression of *Sox9* and *Mef2a/Mef2c* regulons and their target genes in enthesis chondrocytes. (**f, h**) Immunofluorescence shows distribution and dense level of *Sox9* and *Mef2a/Mef2c* during different postnatal time. Scale bars, 100 μm.

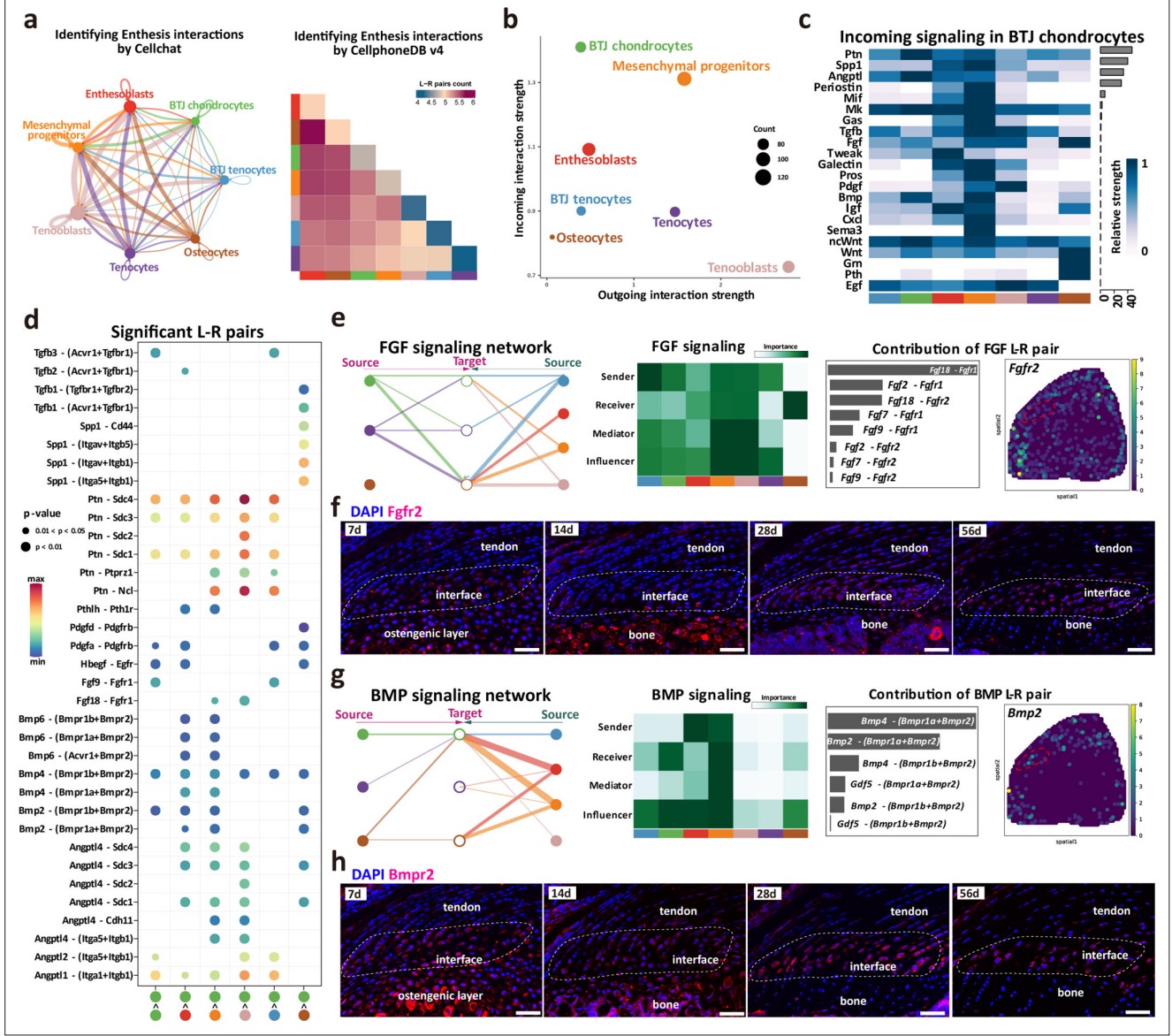

**Figure 6.** Intercellular crosstalk signaling networks regulating enthesis postnatal growth. (**a**) Overview of the cellular network regulating the postnatal enthesis growth predicted by CellChat and Cellphone DB. (**b**) The dominant senders (sources) and receivers (targets) among seven cell clusters. (**c**) Identify signals contributing most to the incoming signaling of bone-tendon junction (BTJ) chondrocytes. (**d**) Heatmap shows the most significant signaling networks among each subcluster. (**e, g**) Overview of FGF and BMP signalings networks in enthesis development. Hierarchy plots show the inferred signaling networks among all cell clusters. Heatmaps show the signaling roles of cell groups. Bar plots show the ligand-receptor pairs contributed significantly to BTJ chondrocytes. Feature plot shows the validation of *Bmpr2* and *Fgfr2* in spatial transcriptomic data. (**f, h**) Validations Fgfr2 and Bmpr2 protein in enthesis by immunofluorescence stainings. Scale bars, 100 μm.

The online version of this article includes the following figure supplement(s) for figure 6:

**Figure supplement 1.** Intercellular crosstalk signaling networks regulating enthesis postnatal growth, related to *Figure 6*.

enthesoblasts to BTJ chondrocytes, suggesting the important role of BMP signaling in enthesis differentiation (*Figure 6—figure supplement 1*). In addition, the communication network of TGF-β and PTH signaling pathways was checked, as BTJ chondrocytes majorly received the stimulation by *TGF-β1*, *TGF-β3*, and *Pthlh*, which had been reported positive in chondrocyte differentiation and biomineralization, respectively (*Bobzin et al., 2021*; *Xiao et al., 2022*; *Figure 6—figure supplement 1*).

## Discussion

Deciphering how a complex enthesis is formed from fetal-like into mature status may shed light on the strategies for enhanced BTJ healing (*Derwin et al., 2018*; *Xiao et al., 2022*). However, the cellular complexity and heterogeneity of developing RC enthesis are poorly understood, as previous studies could hardly resolve it at the single-cell level (*Zhang et al., 2022*). So far, there is only one transcriptomic study for embryonic mouse enthesis has been carried out using single-cell RNA sequencing (*Kult et al., 2021*). However, the development of fibrocartilage at the enthesis of mouse RC occurs no earlier than 2 weeks after birth (*Galatz et al., 2007*), suggesting the investigation at the postnatal stage of cellular and genomic mechanisms in enthesis development is needed. According to the works performed by Fang F et al., (*Fang et al., 2022*), they found enthesis progenitors (*Gli1+* progenitors) and validated their stemness in-vitro and in-vivo, within the timepoints from P11 to P56. In this study, we applied single-cell transcriptome analysis to delineate the comprehensive postnatal enthesis growth with temporal atlas from as early as postnatal day 1 up to postnatal day 56. We next used the spatial transcriptome sequencing on postnatal day-1 mice enthesis to verify the anatomical position of different cell populations. This study may facilitate a better understanding of the enthesis development and add to the single-cell datasets repository of enthesis.

According to prior studies, three distinct populations appear where supraspinatus tendon attaches to the humeral head cartilage: tendon midsubstance progenitors, enthesis progenitors, and primary cartilage progenitors (*Blitz et al., 2013*; *Dyment et al., 2015*). Enthesis morphogenesis involves predominately enthesis progenitors transforming into fibrocartilage, during which process enthesis progenitors are organized as an unmineralized cartilaginous attachment unit and then mineralizes via endochondral ossification postnatally (*Galatz et al., 2007*). We used H&E staining to characterize the morphological changes of the cellular components of enthesis after birth. We found that fibrocartilage did not appear at the BTJ site in mice until 2–3 weeks after birth, and the statistical results showed that enthesis cells' size increased significantly during postnatal development at 7–14 days after birth (*Figure 1d*). At 14 days after birth, these cells showed a typical chondrocyte phenotype. This is consistent with the findings of *Galatz et al., 2007*. According to the literature, the development of fibrocartilage in the enthesis occurs postnatally, which is not evident until 2 weeks after birth (*Galatz et al., 2007*). We consider that the postnatal 7–14 days were the critical stages for fibrocartilage cell differentiation in enthesis, based on the results of our cell morphology studies. Although there is no qualitative evidence for cells with specific gene markers, morphological changes can reflect changes in cell composition and function, which is a very important characteristic change in the study of the development of cell populations. Consistent with the timing we found, Schwartz AG et al., reported that Gli1-expressing cells significantly increased and populated at the enthesis site since postnatal day 7, well before the onset of mineralization, and persisted in the mature enthesis (*Schwartz et al., 2015*). *Gli1*[+] cells and their progenies are retained in the enthesis region throughout postnatal development, contributing hugely to enthesis growth (*Felsenthal et al., 2018*; *Jensen et al., 2018*; *Schwartz et al., 2015*). Fei F et al., also focused on *Gli+* progenitor cells, their data provided important clues to questions related to the development of the enthesis. In order to obtain sequencing data over a broader time span, we combined our own data with that of Fei F et al. And, we further mapped BTJ chondrocytes onto spatial positions by spatial transcriptome sequencing and found that they were consistent with the anatomical positions of enthesis, which verifies the accuracy of our definition of cell populations.

It is generally recognized that difficulties in restoring the mechanical properties after BTJ injury are largely due to the failure of fibrocartilage recapitulation (*Shengnan et al., 2021*). Moreover, in modern RC reconstruction surgeries, anchors fix the tendon to the insertion areas without access to bone marrow, which means BMSCs are not likely the main stem cell source for repair (*Bi et al., 2007*; *Schwartz et al., 2015*; *Utsunomiya et al., 2013*). Therefore, it is needed to investigate the native cellular origination and molecular biology of fibrocartilage formation, in order to enlighten developmental engineering strategies. To investigate the trajectory of postnatal bone-tendon junction cell development. Cellrank analysis was performed, and results showed that the directionality of differentiation between BTJ chondrocytes, tenocytes, and two subsets of osteocytes were independent of each other. The fate probabilities analysis, consistent with RNA velocity, showed that only BTJ chondrocytes contributed to the fate of enthesis chondrocytes (*Figure 3e and f*), suggesting that the fibrocartilage in postnatal enthesis origin from enthesis site-specific progenitors, instead of

tendon cells. We also noticed that *Tnn* was one of the driving genes in chondrogenic fibroblasts differentiating into fibrochondrocytes, and we confirmed the existence of the tenascin N protein (also named tenascin W) in the developing fibrocartilage layer. We still know very little about the basic biology of tenascin W, which has been reported to be expressed in developing and mature bone, specifically in a subset of stem cell niches (*Meloty-Kapella et al., 2006*). Details are scarce, but the stimulating effects of tenascin-W on osteoblastic progenitors' differentiation and migration have been reported (*Meloty-Kapella et al., 2008*; *Morgan et al., 2011*), suggesting its potential role in facilitating enthesis progenitors differentiating into fibrochondrocytes.

To investigate the key factors that regulate enthesis development, CellChat analysis was performed. We found that enthesis cells mainly received signals from other cell types, instead of sending signaling factors. And, we focused on the growth factors signaling pathways that were involved in the enthesis cells network, mainly including the previously reported FGF family (*Bobzin et al., 2021*; *Roberts et al., 2019*), BMP family (*Blitz et al., 2013*), TGF-β family (*Tan et al., 2020*; *Xiao et al., 2022*), and PTH family (*Felsenthal et al., 2018*; *Schwartz et al., 2015*; *Figure 6* and *Figure 6—figure supplement 1*). Among them, we first found that FGF signaling was widely expressed in enthesis cells. As previously mentioned, the development of fibrocartilage in enthesis is similar to a growth plate with an endochondral-like zone. According to the literature, pre-hypertrophic cells in the growth plate express high levels of *Fgfr3*, and hypertrophic chondrocytes express high levels of *Fgfr1* (*Ornitz and Itoh, 2015*). Yet we found enthesis cells mainly expressed *Fgfr2*, suggesting that the regulation of FGFs in enthesis progenitors differentiating into fibrocartilage cells was different from growth plate cartilage. Recent work has confirmed that enthesis development in the mouse mandible was regulated by FGF signaling via FGFR2-FGF2 signaling (*Roberts et al., 2019*). These findings suggest a potential role of FGFR2 in enthesis cells differentiating into fibrocartilage during enthesis development. We next noticed the BMP signaling (specifically Bmp2, Bmp4) was expressed in enthesis cells, as one key feature of the *Sox9* and *Scx* positive progenitors is their dependence on *Bmp2* and *Bmp4* for specification and differentiation (*Blitz et al., 2013*; *Bobzin et al., 2021*). Blitz et al., found that BMP4 derived from the tendon tip induces enthesis progenitors differentiating into chondrocytes, and conditional inactivation of *Bmp4* using *Scx*-Cre blocks formation of the cartilage anlage prefiguring the bone eminence (*Blitz et al., 2009*). Interestingly, it has been indicated that BMP2 could upregulate tenascin-N expression through a p38-dependent signaling pathway (*Scherberich et al., 2005*), and we found *Tnn* was among the top driving genes in fibrochondrocyte differentiation. We also found the TGF-β signaling in enthesis cells, as TGF-β was important due to its crucial role in cartilage and tendon development (*Killian, 2022*). Canonical TGF-β ligands may be diffuse into the near tendon and enthesis, positive in recruiting chondrogenic cells, and the secretion of TGF-β1 has been confirmed mechanically mediated (*Subramanian et al., 2018*; *Xiao et al., 2022*).

There are some certain limitations in the current study. First, we could not remove all the humeral head cells because the enthesis tissues only contain 4–5 layers of cells and are located adjunct to bone marrow and growth plate. Future use of high-precision microdissection approaches to isolate region-specific cells will address the limitation. Second, it is undeniable that spatial transcriptomics are better reliable to address possible dissociation artifacts and gain spatial information. However, utilization of spatial transcriptomic sequencing on enthesis is limited owing to the difficulty in sectioning without decalcification, which restricted our attempt to acquire spatial transcriptomics on mature enthesis with tough bony tissue. Finally, this study was performed predominately on single-cell RNA and spatial transcriptomics datasets, despite we verified the molecules inferred by the analysis algorithm, the whole study was designed as descriptive research, Future studies will label the markers found in this study on transgenic mice and investigate their in-vivo function in enthesis development.

In summary, our study deciphered the cellular complexity and heterogeneity of postnatal enthesis growth by providing descriptive single-cell transcriptomic and spatial datasets. We then revealed the molecular dynamics during fibrocartilage differentiation, providing a valuable resource for further investigation of tendon enthesis development at the mechanistic level, which may facilitate a better understand of the enthesis development and add to the single-cell datasets repository of the enthesis.

## Materials and methods

### Collection of cells from the supraspinatus tendon enthesis

All animal experimental protocols were approved by the Animal Ethics Committee of Central South University (No. 2022020058). The humeral head- supraspinatus tendon samples were dissected from the left shoulders of C57/BL6 mice at postnatal day-1, day-7, day-14, and day-28. In general, samples were harvested from pooled sibling limbs of two litters (five to six limbs per pool). Following dissection, the humeral heads and tendons were trimmed to retain the enthesis part, and all the samples were minced immediately and digested in type I collagenase (1 mg/ml, Gibico) and type II collagenase (1 mg/ml, Gibico) diluted in low-glucose DMEM (Gibco) solution at 37 °C for 30–40 min. Freshly isolated cells were resuspended into FACS buffer containing 2% FBS (Gbico) in PBS. Cell suspensions were stained with antibodies including Ter119-Alexa700 and Cd45- Alexa700 (Biolengend) to remove blood cells. DAPI (BD) stain was used to exclude dead cells. Flow cytometry was performed on BD FACS Aria II, single cells were gated using doublet-discrimination parameters and collected in FACS buffer.

### Single-cell spatial transcriptomic sequencing by stereo-seq

Single-cell spatial transcriptomic sequencing was performed on the BGI stereo-seq platform (*Chen et al., 2022*). Briefly, the tissue section of the postnatal day-1 left shoulder of C57/bl6 mice was placed on the Stereo-seq chip (1 cm * 1 cm), then incubated and stained with a mixture of nucleic acid reagent (Invitrogen, Q10212). Section images were captured using a Zeiss Axio Scan Z1 microscope (at EGFP wavelength, 10 ms exposure). Tissue sections were then permeated to release RNAs from the permeated tissue and captured by a Stereo-seq chip. RNAs were then reverse transcribed and the cDNA-containing chips were then amplified with Hot Start DNA Polymerase (QIAGEN). In the library preparing procedure, samples were tagmented with Tn5 transposases (Vazyme) and amplified. After amplification, the PCR products were used for DNB (DNA Nano Ball) generation. Finally, the DNBs were sequenced on the DNBSEQTM T10 sequencing platform (MGI, Shenzhen, China).

### Spatial mapping of cell states with cell2location

Cell2location was used to deconvolute and map single-cell clusters onto spatial transcriptomics spots. In brief, we first estimated reference signatures of cell states using scRNA-seq data of each region and a negative binomial regression model provided in the cell2ocation package (*Kleshchevnikov et al., 2022*). The regression model for the single-cell data was initialized with default settings. The model was then trained using a maximum of 30,000 epochs. The inferred reference cell type signatures were used for cell2location cell-type mapping for corresponding regions that estimate the abundance of each cell state in each spot.

### Droplet-based scRNA-seq

8000–10,000 cells were loaded for each age group by Chromium instrument and its chemistry kit V3 (10 X Genomics) according to the manufacturer's guidance. Each cell was encapsulated with a barcoded Gel Bead in a single partition, then amplified to generate single-cell cDNA libraries and sequenced on an Illumina NovaSeq 6000 platform at a sequencing depth of ~500 million reads. The Cellranger pipeline (version 6.1.1) was used to align the raw reads to the mouse reference genome GRCm38 and to generate feature-barcode matrices. All the low-quality reads were filtered with default parameters.

### Single-cell data processing, quality control, and integration

All the feature-barcode matrices were loaded by the Seurat package (v4.1.0) (*Hao et al., 2021*), doublets or cells with poor quality were removed (less than 200 genes and greater than 2 Median absolute deviations above the median, or more than 5% genes mapping to the mitochondrial genome). After quality control, all the feature data were scaled with the sctransform algorithm, to avoid unwanted variation including percentages of mitochondrial reads, number of detected genes, and predicted cell cycle phase effect. All the datasets were integrated and batch-corrected by using SCVI with default parameters. Furthermore, this integrated data was analyzed and subclustered to

exclude uninterested clusters (including immune cells, red blood cells, endothelial cells, smooth muscle cells, and neural cells).

## Dimensionality reduction, clustering, and DEGs analysis

We used the Uniform Manifold Approximation and Projection (UMAP) and Potential of Heat diffusion for Affinity-based Transition Embedding (PHATE) (*Moon et al., 2019*) method to visualize the dataset in low dimensions. Furthermore, the K-nearest neighbor (KNN) method and the Louvain algorithm were applied to cluster the cells, with 50 PCs selected and resolution set to 2.4, resulting in nine major cell clusters for subsequent analyses. For second-round chondrocyte sub-clustering, we reconstructed the SNN graphs for BTJ clusters, and three subclusters were determined resolution set as 0.6 for each fibrochondrocyte cluster. The FindAllMarkers function in Seurat was used to calculate DEGs among different clusters, the 'test.use' function was set to a statistical framework called MAST (*Finak et al., 2015*). Genes met the criteria that (1) expressing a minimum fraction of 10% in either of the two tested populations; (2) at least a 0.1-fold difference (log-scale) between the two tested populations; (3) adjusted p values less than 0.01, were considered as signature genes. Clusters were annotated according to the expression of those highly variable genes reported in the literature.

## Time-dependent gene signature clustering

DEGs between different timepoints were acquired FindAllMarkers function in Seurat, temporal pattern analysis, and visualization were conducted by using R package Tcseq according to a standard pipeline. Through the above algorithm, the time-dependent DEGs were divided.

## Trajectory analysis and cell state analysis

Before trajectory analysis, the S4 Seurat object was transformed into an anndata object using the Seuratdisk package and loaded by Scanpy (*Wolf et al., 2018*). Then, all the bam files were processed with Velocyto (*La Manno et al., 2018*) to quantify the spliced and unspliced mRNA counts. Subsequently, the Velocyto outputs were loaded into scVelo (*Bergen et al., 2020*) and merged with the anndata object from Scanpy to compute RNA velocity vectors. Low abundance genes (less than 30 total counts) were filtered from the merged dataset. After RNA velocity analysis, we used Cellrank (*Lange et al., 2022*) package to compute infer the terminal cell state and cluster absorption probabilities using nearest-neighbour relationships and RNA velocity with equal weight in CellRank's Markov chain model.

## Cell-cell interaction analysis

Cell-cell interaction analysis was performed using CellChat (*Jin et al., 2021*) package, according to a standard pipeline.

## Gene regulatory network analyses

We applied Single-Cell Regulatory Network Inference and Clustering (SCENIC) (*Aibar et al., 2017*) to identify the cluster-specific gene regulatory networks. The pySCENIC grn method was performed for building the initial co-expression gene regulatory networks (GRN). The regulon data was then analyzed using the RcisTarget package to create TF motifs referring to the mm9-tss-centered-10kb-7 database. The regulon activity scores were calculated using the Area Under the Curve (AUC) method. Besides, we used Cellcall package to analyze the cluster-specific TF enrichment and intercellular communication by combining the expression of ligands/receptors and downstream TF activities for certain L-R pairs. Genes that were expressed in less than 10% of the cells of a certain cell type were excluded.

## GO enrichment analysis

GO enrichment of cluster differentially expressed genes was performed using the R package clusterProfiler (*Wu et al., 2021*), with a Benjamini–Hochberg (BH) multiple testing adjustment and a false-discovery rate (FDR) cutoff of 0.1. The Gene Ontology Resource database (http://geneontology.org) was used for GO pathway analysis. Module scores for each gene set were calculated using the AddModuleScore function implemented in Seurat. Gene sets used for scoring (Chondrocyte proliferation, Proteoglycans synthesis, Cartilage homeostasis, Collagen synthesis, Regulation of bone development, Biomineralization, Negative regulation of bone mineralization) were selected from the Gene

Ontology Browser of MGI Database (C5: biological process gene sets). Visualization was performed using the R package ggplot2.

## Sample harvest and histological observation

The left shoulder of the C57/BL6 mice was harvested on postnatal days 1, 3, 7, 14, 28, and 56. Specimens were obtained and fixed with 10% formalin buffer for 24 hr and rinsed by dual evaporated water then gradually dehydrated by sequential immersion in 70%, 80%, 90%, and 100% alcohol (each for 2 hr), finally dried in the air before use. The samples were embedded in paraffin and then sectioned for histological studies with H-E and Toluidine blue/Fast green staining. Histologic sections were observed using light microscopy (CX31, Olympus, Germany).

## Immunofluorescence staining

The left shoulder of the C57/BL6 mice was harvested on postnatal days 1, 3, 7, 14, 28, and 56. Then fixed with 4% neutral buffered formalin for 24 hr. After decalcifying, dehydrating, and embedded in OCT, specimens were longitudinally sectioned with 10 μm. For immunofluorescence staining, the sections were washed with PBS, permeabilized with 0.1% TritonX-100, and then blocked with 5% bovine serum albumin (BSA; Sigma-Aldrich). Sections were incubated with primary antibodies anti-Sox9 (Abcam, ab185996), anti-Scx (Santa Cruz, sc518082), anti-Tnn (Thermo Fisher, ab_2900654), anti-Mef2a/Mef2c (Abclonal, A2710), and anti-Mfge8 (Abclonal, A12322) at 4 °C overnight, then incubated with Alexa-Fluor 488 conjugated secondary antibody (Abcam, ab150129) and Alexa-Fluor 594(Abcam, ab150120) conjugated secondary antibody at room temperature for 1 hr and counterstained with DAPI (Invitrogen, USA). All the images were observed and captured using a Zeiss Axio-Imager.M2 microscope (Zeiss, Germany) equipped with an Apotome.2 System. Densities of Sox9, Scx, Tnn, Mef2a/Mef2c, or Mfge8 positive cells of each captured image were measured using 200 x magnification graphs for each slide by the Image J software (National Institutes of Health, Bethesda, MD). The antibodies used in this study were listed in *Supplementary file 1*.

## Statistical analysis

One-way ANOVA and Student's t-test were performed to assess whether there were statistically significant differences in the results between time groups. Values of $p < 0.05$ were considered to be significantly different. Data were analyzed using Prism 7 software (GraphPad).

## Acknowledgements

The authors would like to thank Professor Hui Xie, Xiang-Hang Luo, and other staff from the Movement System Injury and Repair Research Center, Xiangya Hospital, Central South University, Changsha, China, for their kind assistance during the experiments. Funding this study was supported by the National Natural Science Foundation of China (NO. 82230085, and 82272572).

## Additional information

### Funding

| Funder | Grant reference number | Author |
| --- | --- | --- |
| National Natural Science Foundation of China | 82230085 | Hongbin Lu |
| National Natural Science Foundation of China | 82272572 | Hongbin Lu |

The funders had no role in study design, data collection and interpretation, or the decision to submit the work for publication.

### Author contributions

Tao Zhang, Conceptualization, Data curation, Software, Formal analysis, Validation, Investigation, Visualization, Methodology, Writing – original draft; Liyang Wan, Visualization, Methodology, Writing

– original draft; Han Xiao, Data curation, Formal analysis; Linfeng Wang, Data curation, Validation; Jianzhong Hu, Conceptualization, Supervision, Project administration; Hongbin Lu, Conceptualization, Supervision, Funding acquisition, Project administration

### Author ORCIDs
Tao Zhang http://orcid.org/0000-0002-6746-8834
Liyang Wan http://orcid.org/0000-0003-2194-1080
Hongbin Lu http://orcid.org/0000-0001-7749-3593

### Ethics
All animal experimental protocols were approved by the Animal Ethics Committee of Central South University (No. 2022020058).

### Decision letter and Author response
Decision letter https://doi.org/10.7554/eLife.85873.sa1
Author response https://doi.org/10.7554/eLife.85873.sa2

## Additional files

### Supplementary files
• Supplementary file 1. Table Antibodies used in this study.
• Supplementary file 2. Filtered out genes, related to *Figure 2*.
• Supplementary file 3. Cluster markers, related to *Figure 2*.
• Supplementary file 4. Enthesis chondrocyte time-dependent markers, related to *Figure 4*.
• Supplementary file 5. Monocle3 gene module signature, related to *Figure 4*.
• Supplementary file 6. Pyscenic topRegulators, related to *Figure 5*.
• MDAR checklist

### Data availability
All single-cell datasets created during this study are publicly available at the Gene Expression Omnibus (GSE223751). Any additional information required to re-analyze the data in the paper is available from the corresponding author upon request.

The following dataset was generated:

| Author(s) | Year | Dataset title | Dataset URL | Database and Identifier |
|---|---|---|---|---|
| Zhang T, Wan LY | 2022 | Single-cell RNA sequencing reveals cellular and molecular heterogeneity in fibrocartilaginous enthesis formation | https://www.ncbi.nlm.nih.gov/geo/query/acc.cgi?acc=GSE223751 | NCBI Gene Expression Omnibus, GSE223751 |

The following previously published dataset was used:

| Author(s) | Year | Dataset title | Dataset URL | Database and Identifier |
|---|---|---|---|---|
| Fang F, Thomopoulos S | 2022 | Single-cell RNA-seq of the tendon enthesis cells | https://www.ncbi.nlm.nih.gov/geo/query/acc.cgi?acc=GSE182997 | NCBI Gene Expression Omnibus, GSE182997 |

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
