## [Editor Report]

This paper represents a valuable single-cell level analysis of tendon enthesis development. The study allows further understanding of this specific process with clinical implications. The authors provided convincing evidence for the heterogeneity of postnatal enthesis growth and the molecular dynamics and signaling networks during enthesis formation.

---

## [Decision Letter]

**Decision letter after peer review:**

Thank you for submitting your article "Single-cell RNA sequencing reveals cellular and molecular heterogeneity in fibrocartilaginous enthesis formation" for consideration by *eLife*. Your article has been reviewed by 3 peer reviewers, and the evaluation has been overseen by a Reviewing Editor and Murim Choi as the Senior Editor. The following individual involved in the review of your submission has agreed to reveal their identity: Xiao Chen (Reviewer #3).

Essential revisions:

Although the editors and reviewers agreed that the work is valuable, it needs further improvements to warrant publication in *eLife*. Major issues are:

1) The paper should be discussed relative to the results in Fang et al., Cell Stem Cell, 2022.

2) The importance of P7 as a critical differentiation timepoint is not well supported.

3) Validation is needed (e.g., IHC and/or FISH) for many of the scRNAseq results (e.g., CellChat pathways).

*Reviewer #1 (Recommendations for the authors):*

1. The results and their novelty should be discussed in comparison to the recent Cell Stem Cell study describing enthesis development using scRNAseq and lineage tracing approaches (https://doi.org/10.1016/j.stem.2022.11.007).

2. Figure 1d: The PCA for histomorphologic parameters (which typically have high variations) does not show any meaningful separations or groupings. I do not agree with the authors' conclusion that this analysis reveals P7 as a critical developmental timepoint. In general, PCA may not be appropriate for this data set.

3. According to the methods, it appears that the entire humeral head-supraspinatus tendon was used for cell isolation for scRNAseq. This results in the inclusion of cells from bone, growth plate, enthesis and tendon. As such, only a very small percentage of cells came from the enthesis, as is clear from the cell clusters in Figure 2b and 2c. This is a flaw in the approach; inclusion of such a wide range of cells makes interpretation of "enthesis" cells difficult, as described in more detail in the comments below.

4. The differentiation/pseudotime analysis described in Figure 3 is difficult to follow. I do not think it is useful to combine cell transcriptomes from vastly different tissues and then define a velocity map. There is too much varied information for the algorithm to create valid connections, as the there will be many many branches/paths from mesenchymal stem cell to osteoblast, tenocyte, chondrocyte, etc. Presumably, embedded in these maps are trajectories for osteoblast differentiation, chondrocyte differentiation, tenocyte differentiation, etc. There are too many layers of overlapping information to deduce anything meaningful for the small number of cells associated with the enthesis.

5. The authors uses the term "function" throughout the paper (e.g., "functional definition of fibrocartilage subpopulations"). However, this is a descriptive study, and "function" (or mechanism) can only be theoretically inferred from the various algorithms used to analyze the data. A role for any of the pathways or processes can only be defined with gain- and/or loss-of-function studies.

6. "C2 highly expressed biomineralization-related genes (Clec3a, Tnn, Acan)". The three example genes are not related to biomineralization.

7. The functional characterization of the three enthesis cell clusters is not convincing. For example, activation of metabolism-related processes is a vague result than can mean a lot of things (including changes in differentiation), yet the authors interpret it very specifically as " role in postnatal fibrochondrocyte formation and growth".

8. The pseudotime analysis of the enthesis cell clusters is not convincing. The three clusters are quite close and overlapping on the UMAP. Furthermore, the authors focus on Tnn as a novel and unique gene, yet the expression pattern shown in Figure 5g implies even expression of this gene across all three clusters.

9. The TC1 markers (Ly6a, Dlk3, Clec3b) imply a non-tendon-specific cell population. Perhaps a tendon progenitor pool or an endothelial cell phenotype is more appropriate.

10. Pseudotime analyses assume that your data set includes cells from progenitor through mature cell populations. It is unclear that the timepoints studied here included cells from early progenitor states.

11. The CellChat analysis is not useful, as the authors included 18 cell types. The number of possible interactions among so many cell types is enormous, and deducing valid connections between any two cell types in this case is questionable.

12. The authors should validate their key scRNAseq results with in situ hybridization. Only a single gene, Tpp, was validated on tissue sections. This validation is particularly important for this study because the authors included a wide variety of tissues/cells in their isolation and analyses.

13. The authors should demonstrate functional necessity of at least one gene/pathway identified by the scRNAseq analyses (e.g., through gene knockout).

*Reviewer #2 (Recommendations for the authors):*

1. As known, Fei Fang et al. have established single-cell transcriptomes of mouse supraspinatus tendon enthesis cells (Cell Stem Cell, 2022). It is suggested that the authors introduced Fei Fang et al.'s work in Introduction and emphasize the significant novelty compared with Fei Fang et al.'s work.

2. In Figure 1, the authors highlighted P7 was a critical stage for enthesis differentiation. But this section was less associated with the following content. The authors should link these results with the scRNASeq data. Is there any time-dependent change/signaling in scRNASeq data at this critical time point?

3. In the H and E staining of Figure 1A, the tendon structure was separated and random. It is suggested that the authors provide high-quality staining figures.

4. Figure 2 showed that the Scx+ or *Sox9*+ cells was decreased in enthesis over time. At least it should be co-staining to show the distribution and frequency of double positive and single positive cell populations. However, a previous study has demonstrated this finding (PLOS ONE, 2020). It is suggested to verify some new findings by IF or IHC staining.

5. There are some conflicts about trajectory analysis. In Figure 3C, RNA velocity showed that the arrow flowed from BTJ to MTJ and CTFb. However, in Fig3d, PAGA plot indicated that BTJ cells is independent of other cells. Furthermore, in supplementary figure S3, RNA velocity showed that the trajectory flowed from TC to BTJ. These figures were inconsistent with the described results. Please provide detailed explanation to avoid misleading readers.

6. Figure 5 showed that C1 was the original cluster, and whether C1 cluster expressed canonical progenic/stem cell markers.

7. The authors performed cell-cell interaction based on cellchat analysis. But the cell-cell interaction was not actively examined.

*Reviewer #3 (Recommendations for the authors):*

1. Fang et al. (A mineralizing pool of Gli1-expressing progenitors builds the tendon enthesis and demonstrates therapeutic potential. Cell stem cell. 2022) defined enthesis cell transcriptomes and differentiation trajectories, and identified Gli1+ progenitor population for enthesis. Please further clarify the innovation of your research, and in depth introduction or discussion is needed to compare and contrast the results between the two research.

2. In Figure 1, more evidence are needed to prove that Neonatal to postnatal day 7 (P7) is the critical stage for enthesis fibrocartilage cell differentiation, for example, immunofluorescence staining or qPCR for enthesis fibrocartilage cell makers, instead of relying on H and E only.

3. Line123. Figure 2e showed that the expression of Clec3a and Col2a1 were low in c4," which were ubiquitously expressed in bone-tendon junction cell (c4)" seems to be an inexact expression.

4. Line 117, which cell clusters belong to "fibroblast-associated cells"?

5. Line 125, it is better to co-staining the scx and *Sox9* to validate the existence of BTJ cells. Scx and *Sox9* are known markers of BTJ, do you have find new makers for BTJ by scRNA-seq?

6. Line 148, "stemness" degree? Are there other evidence, such as stem cell maker expression, to show that "growth plate cells and fibroblasts associated clusters are higher than other cell types". The expression of "stemness" seems exaggerated.

7. There is no description of figure 4b in the results.

8. In figure 5, 2-3 makers identified by scRNA-seq for fibrocartilage formation are suggested to be validated by immunofluorescence stainning or other methods, instead of only proving the Tnn expression in postnatal BTJ growth.

9. There are no verification of the signaling network for the enthesis postnatal growth which were revealed by Cellchat. It is suggested to validate the key signaling, such as Bmpr2 signaling.

---

## [Author Response]

Essential revisions:Although the editors and reviewers agreed that the work is valuable, it needs further improvements to warrant publication in eLife. Major issues are:1) The paper should be discussed relative to the results in Fang et al., Cell Stem Cell, 2022.2) The importance of P7 as a critical differentiation timepoint is not well supported.3) Validation is needed (e.g., IHC and/or FISH) for many of the scRNAseq results (e.g., CellChat pathways).

Thank you for your kind suggestions. According to your comments, we carefully revised the paper, and the issues mentioned in prior version has been addressed:

(1) We discussed the relation and consistency between our dataset and Fang et al. in writing. Meanwhile, we integrated our dataset with open source GSE182997 datasets (3 sample) provided by Fang et al. Consequently, we are now better able to discuss the enthesis development from P1 to P58, including 7 timepoints to ideally cover the whole postnatal enthesis growth.

(2) We removed the statement that P7 was a critical differentiation timepoint in revised paper. And we furtherly added toluidine blue staining to observe the extracellular matrix of fibrocartilage. At the same time, we performed immunochemistry to test the protein level of collagen II at enthesis.

(3) We performed immunofluorescence staining to validate the results from cell trajectory inference (Tnn and Mfge8 expression, Figure 4), transcription factor calculation (Mef2a/2c and *Sox9* expression, Figure 5), and communication results (Fgf2r and Bmp2r expression, Figure 6). And we performed spatial transcriptome sequencing on enthesis slide at postnatal day 1 to exam the location of these results.

Reviewer #1 (Recommendations for the authors):1. The results and their novelty should be discussed in comparison to the recent Cell Stem Cell study describing enthesis development using scRNAseq and lineage tracing approaches (https://doi.org/10.1016/j.stem.2022.11.007).

Thank you for your suggestions. We discussed the relation and consistency between our dataset and Fang et al. (revised in Figure 2, Figure 2—figure supplement 2 and lines 75-79). Meanwhile, we integrated our dataset with open source GSE182997 datasets (3 sample) provided by Fang F et al. Consequently, we are now better able to discuss the enthesis development from P1 to P58, including 7 timepoints to ideally cover the whole postnatal enthesis growth.

2. Figure 1d: The PCA for histomorphologic parameters (which typically have high variations) does not show any meaningful separations or groupings. I do not agree with the authors' conclusion that this analysis reveals P7 as a critical developmental timepoint. In general, PCA may not be appropriate for this data set.

Thank you for your suggestions. We removed the PCA results in revised paper. And we realized the improper statement about P7 as a critical developmental timepoint. We furtherly added toluidine blue staining to observe the extracellular matrix of fibrocartilage. At the same time, we performed immunochemistry to test the protein level of collagen II at enthesis.

We found fibrocartilage was not evident at the enthesis of mouse rotator cuff until 2-3 weeks after birth, as evidenced by toluidine blue stained cartilage ECM and IHC validated Col2a1 expression. Yet we found the cell sizes of enthesis cells remarkably increased during postnatal day 7-14, and the enthesis cells were observed as typical chondrocyte phenotype, column-like stacked alongside the direction of tendon fiber. These findings led us the conclusion that the development of fibrocartilage in the enthesis occurs postnatally. (revised in Figure 1 and lines 94-104)

3. According to the methods, it appears that the entire humeral head-supraspinatus tendon was used for cell isolation for scRNAseq. This results in the inclusion of cells from bone, growth plate, enthesis and tendon. As such, only a very small percentage of cells came from the enthesis, as is clear from the cell clusters in Figure 2b and 2c. This is a flaw in the approach; inclusion of such a wide range of cells makes interpretation of "enthesis" cells difficult, as described in more detail in the comments below.

Thank you for your suggestions, and we are sorry for the misinterpretation about sample preparation in prior manuscription.

In our practice, the humeral head- supraspinatus tendon samples were dissected from the left shoulders of C57/BL6 mice at postnatal day-1, day-7, day-14, and day-28. In general, samples were harvested from pooled sibling limbs of two litters (five to six limbs per pool). Following dissection, the humeral heads and tendons were trimmed to retain the enthesis part, and all the samples were minced immediately and digested. (revised in Lines 456-458)

Although, we can’t deny the fact that we could not remove all the humeral head cells under microscope because the enthesis tissues only contained 4-5 layers of cells and located adjunct to bone marrow and growth plate. The enthesis datasets inevitably contains some cells came from bone marrow and growth plate in humeral head, as shown by the work reported by Fang F et al. (Figure 1a, Cell Stem Cell. 2022) (revised in Lines 406-409)

In order to better understand the preciseness of our cluster and annotation results, we furtherly performed single cell spatial transcriptomic sequencing on postnatal day 1 mice enthesis ice section, we then deconvoluted and mapped P1 single-cell clusters (BTJ chondrocytes) onto spatial transcriptomics spots, and the cell2location results showed that BTJ chondrocytes we had annotated in single cell dataset could be properly mapped onto enthesis site (Figure 2e). (revised in Lines 134-136)

However, utilization of spatial transcriptomic sequencing on enthesis are limited owing to the difficulty in sectioning without decalcification, which restricted our attempt to acquire spatial transcriptomics on mature enthesis with tough bony tissue. (revised in Lines 409-413)

4. The differentiation/pseudotime analysis described in Figure 3 is difficult to follow. I do not think it is useful to combine cell transcriptomes from vastly different tissues and then define a velocity map. There is too much varied information for the algorithm to create valid connections, as the there will be many many branches/paths from mesenchymal stem cell to osteoblast, tenocyte, chondrocyte, etc. Presumably, embedded in these maps are trajectories for osteoblast differentiation, chondrocyte differentiation, tenocyte differentiation, etc. There are too many layers of overlapping information to deduce anything meaningful for the small number of cells associated with the enthesis.

Thank you for your suggestions. We realized the bad effects caused by too many layers of overlapping information in our prior results, and we re-analyzed the differentiation pseudotime across cell types.

Because we integrated our dataset with open source GSE182997 datasets (3 sample) provided by Fang et al., thus we re-clustered and re-annotated the cell clusters. We first performed Cytotrace and Cellrank analysis across all cell clusters to predict cellular differential potential and terminal differentiate states. And we identified differentiation trajectories of these 3 terminal cell types (revised in Figure 3 and lines 155-184).

Our next step was performed by refining the input for pseudotime analysis (including mesenchymal progenitors, enthesoblasts, and enthesis chondrocytes), and we focused on BTJ chondrocyte differentiation trajectory to reconstruct the gene dynamics (revised in Figure 4 and lines 197-225).

5. The authors uses the term "function" throughout the paper (e.g., "functional definition of fibrocartilage subpopulations"). However, this is a descriptive study, and "function" (or mechanism) can only be theoretically inferred from the various algorithms used to analyze the data. A role for any of the pathways or processes can only be defined with gain- and/or loss-of-function studies.

Thank you for your suggestions. We realized the improper statement of functionality in the prior version of paper. We scrutinized the paper and corrected the statement. (revised in Figure 5 and lines 241-249)

6. "C2 highly expressed biomineralization-related genes (Clec3a, Tnn, Acan)". The three example genes are not related to biomineralization.

Thank you for your suggestions. In combination with GSE182997 datasets provided by Fang et al., we re-analyzed the clustering outcomes of BTJ chondrocytes and we realized it is inappropriate to subcluster the current BTJ chondrocytes. And we removed the results related to the three enthesis cell subclusters in revised version. At the same time, we recalculated the biomineralization-related genes (*Runx2*, *Ibsp*, *Spp1*, *Col11a1*) in the revised manuscription. (revised in Figure 3h and 4f)

7. The functional characterization of the three enthesis cell clusters is not convincing. For example, activation of metabolism-related processes is a vague result than can mean a lot of things (including changes in differentiation), yet the authors interpret it very specifically as " role in postnatal fibrochondrocyte formation and growth".

Thank you for your suggestions. Relating to question 6, we re-analyzed the clustering outcomes of BTJ chondrocytes and we believed it is inappropriate to subcluster the current BTJ chondrocytes. And we removed the results related to the three enthesis cell subclusters in revised version.

8. The pseudotime analysis of the enthesis cell clusters is not convincing. The three clusters are quite close and overlapping on the UMAP. Furthermore, the authors focus on Tnn as a novel and unique gene, yet the expression pattern shown in Figure 5g implies even expression of this gene across all three clusters.

Thank you for your suggestions. We re-performed differentiation pseudotime analysis (including mesenchymal progenitors, enthesoblasts, and enthesis chondrocytes), and we focused on BTJ chondrocyte differentiation to reconstruct the gene dynamics. (revised in Figure 4 and lines 197-225)

We found the expression of BTJ chondrocytes differentiate driving genes *Mfge8* and *Tnn* had not been reported, and we furtherly validated the expression of *Mfge8* and *Tnn* in enthesis by immunofluorescence staining. (revised in Figure 4)

9. The TC1 markers (Ly6a, Dlk3, Clec3b) imply a non-tendon-specific cell population. Perhaps a tendon progenitor pool or an endothelial cell phenotype is more appropriate.

Thank for your suggestions. We decided not to discuss the tendon fate in this study, in consideration of the tissue digest and cell isolation methods was designed for extracting enthesis chondrocytes, instead of tendon cells. And we have removed the results and discussion in association with tendon subpopulations.

10. Pseudotime analyses assume that your data set includes cells from progenitor through mature cell populations. It is unclear that the timepoints studied here included cells from early progenitor states.

Thank you for your suggestions. According to the works performed by Fang F et al. (Cell Stem Cell. 2022), they found enthesis progenitors (Gli1+ progenitors) and validated their stemness in-vitro and in-vivo, within the timepoints from P11 to P56 (revised in lines 325-328).

From enthesis histological results, enthesis specific fibrochondrocytes are observable no early than postnatal 14 days, suggested that the enthesis chondrocyte formation is a postnatal progress. We integrated our dataset with GSE182997 datasets (3 sample) provided by Fang F et al. Consequently, we discussed the enthesis development from P1 to P58, including 7 timepoints to ideally cover the whole postnatal enthesis growth (revised in lines 324-331).

11. The CellChat analysis is not useful, as the authors included 18 cell types. The number of possible interactions among so many cell types is enormous, and deducing valid connections between any two cell types in this case is questionable.

Thank you very much for your suggestions. We refined the cellchat input for downstream analysis, including 7 clusters of enthesis-related cells (BTJ chondrocytes, BTJ tendons, Tenocytes, Osteocytes, Enthesoblasts, and Mesenchymal progenitors). Then we checked the expression and location of CellChat inferred target genes in spatial transcriptomic data. Meanwhile we performed immunofluorescence staining to check the *Fgfr2* and *Bmpr2* expression inferred by cellchat. (revised in Figure 6 and lines 286-303)

12. The authors should validate their key scRNAseq results with in situ hybridization. Only a single gene, Tpp, was validated on tissue sections. This validation is particularly important for this study because the authors included a wide variety of tissues/cells in their isolation and analyses.

Thank you for your suggestions. validate the results from cell trajectory inference (*Tnn* and *Mfge8* expression, Figure 4), transcription factor calculation (*Mef2a/2c* and *Sox9* expression, Figure 5), and communication results (*Fgf2r* and *Bmp2r* expression, Figure 6). And we performed spatial transcriptome sequencing on enthesis slide at postnatal day 1 to exam the location of these results in enthesis.

13. The authors should demonstrate functional necessity of at least one gene/pathway identified by the scRNAseq analyses (e.g., through gene knockout).

Thank you very much for your suggestions. We aimed to provide a transcriptional resource for further investigation of fibrocartilage development.

We found expression of *Tnn*, *Clec3a*, and *Mfge8* as significant driving genes in enthesis chondrocytes differentiation, suggesting their possibilities as new makers for BTJ chondrocytes, and we checked their expression in spatial transcriptomic data. Furtherly we stained the protein expression of *Tnn* and *Mfge8* to validate their location.

The whole study was designed as descriptive research, Future studies will label the markers found in this study on transgenic mice and investigate their in-vivo function in enthesis development. (revised in Figure 4 and lines 412-415)

Reviewer #2 (Recommendations for the authors):1. As known, Fei Fang et al. have established single-cell transcriptomes of mouse supraspinatus tendon enthesis cells (Cell Stem Cell, 2022). It is suggested that the authors introduced Fei Fang et al.'s work in Introduction and emphasize the significant novelty compared with Fei Fang et al.'s work.

Thank for your suggestions, we discussed the relation and consistency between our dataset and Fang et al. Meanwhile, we integrated our dataset with open source GSE182997 datasets (3 sample) provided by Fang et al. Consequently, we hope to be better able to discuss the enthesis development from P1 to P58, including 7 timepoints to ideally cover the whole postnatal enthesis growth.

2. In Figure 1, the authors highlighted P7 was a critical stage for enthesis differentiation. But this section was less associated with the following content. The authors should link these results with the scRNASeq data. Is there any time-dependent change/signaling in scRNASeq data at this critical time point?

Thank for your suggestions, we removed the PCA results in revised paper. And we furtherly added toluidine blue staining to observe the extracellular matrix of fibrocartilage. At the same time, we performed immunochemistry to test the protein level of collagen II at enthesis.

We found fibrocartilage was not evident at the enthesis of mouse rotator cuff until 2-3 weeks after birth, as evidenced by toluidine blue stained cartilage ECM and IHC validated col2a1 expression. Yet we found the cell sizes of enthesis cells remarkably increased during postnatal day 7-14, and the enthesis cells were observed as typical chondrocyte phenotype, column-like stacked alongside the direction of tendon fiber. These findings led us the conclusion that the development of fibrocartilage in the enthesis occurs postnatally. (revised in Figure 1 and lines 90-104)

And we summarized the time-dependent gene dynamic and bioprocess change in BTJ chondroctyes differentiation. (revised in Figure 4 and lines 197-225)

3. In the H and E staining of Figure 1A, the tendon structure was separated and random. It is suggested that the authors provide high-quality staining figures.

Thank for your suggestions, and we restained the sections with H-E and toluidine blue to better show the structure change in enthesis growth. At the same time, we performed immunochemistry to test the protein level of collagen II at enthesis. (revised in Figure 1a)

4. Figure 2 showed that the Scx+ or Sox9+ cells was decreased in enthesis over time. At least it should be co-staining to show the distribution and frequency of double positive and single positive cell populations. However, a previous study has demonstrated this finding (PLOS ONE, 2020). It is suggested to verify some new findings by IF or IHC staining.

Thank for your suggestions. We co-stained Scx+ and *Sox9*+ and recalculated Scx and *Sox9* positive cells in enthesis. We also checked Scx and *Sox9* expression in our P1 spatial transcriptomic data. (revised in Figure 2e, 2f and lines 136-141)

In differentiation pseudotime analysis, we identified the most highly significant driving genes (Klf9, Fxyd5, Klf4, Mfge8, *Sox9*, Clec3a, Wwp2, Tnn), we then confirmed the expression of Klf9, Klf4, Clec3a, Mfge8, and Tnn in single cell spatial transcriptomics, except for previously reported *Sox9* and Wwp2 which relative to chondrogenesis (Blitz et al., 2013). We found the expression of Mfge8 and Tnn had not been reported, and we validated the expression of Mfge8 and Tnn proteins in enthesis. (revised in Figure 4 and lines 216-225)

5. There are some conflicts about trajectory analysis. In Figure 3C, RNA velocity showed that the arrow flowed from BTJ to MTJ and CTFb. However, in Fig3d, PAGA plot indicated that BTJ cells is independent of other cells. Furthermore, in supplementary figure S3, RNA velocity showed that the trajectory flowed from TC to BTJ. These figures were inconsistent with the described results. Please provide detailed explanation to avoid misleading readers.

Thank you for your suggestions. Because we integrated our dataset with open source GSE182997 datasets (3 sample) provided by Fang et al., thus we re-clustered and re-annotated the cell clusters. Therefore, we re-analyzed the differentiation pseudotime across different cell types.

We first performed Cytotrace and Cellrank analysis across all cell clusters to predict cellular differential potential and terminal differentiate states. And we identified differentiation trajectories of these 3 terminal cell types (revised in Figure 3 and lines 163-175).

Our next step was performed by refining the input for pseudotime analysis (including mesenchymal progenitors, enthesoblasts, and enthesis chondrocytes), and we focused on BTJ chondrocyte differentiation to reconstruct the gene dynamics (revised in Figure 4 and lines 209-225).

6. Figure 5 showed that C1 was the original cluster, and whether C1 cluster expressed canonical progenic/stem cell markers.

Thank you for your suggestions. In combination with GSE182997 datasets (3 sample) provided by Fang et al., we re-analyzed the clustering outcomes of BTJ chondrocytes and we believed it is inappropriate to subcluster the current BTJ chondrocytes. And we removed the results related to the three enthesis cell clusters in revised version.

We compared canonical progenic stem cell markers (Ly6a, Cd34, and Cd44) between BTJ chondrocytes, enthesoblasts and mesenchymal progenitors (revised in Figure 4—figure supplement 1).

7. The authors performed cell-cell interaction based on cellchat analysis. But the cell-cell interaction was not actively examined.

Thank for your suggestions, we refined the cellchat input for downstream analysis, including 7 clusters of enthesis-related cells (BTJ chondrocytes, BTJ tendons, Tenocytes, Osteocytes, Enthesoblasts, and Mesenchymal progenitors). Then we checked the expression and location of CellChat inferred target genes in spatial transcriptomic data. Meanwhile we performed immunofluorescence staining to check the *Fgfr2* and Bmpr2 expression inferred by cellchat. (revised in Figure 6 and lines 286-303)

Reviewer #3 (Recommendations for the authors):1. Fang et al. (A mineralizing pool of Gli1-expressing progenitors builds the tendon enthesis and demonstrates therapeutic potential. Cell stem cell. 2022) defined enthesis cell transcriptomes and differentiation trajectories, and identified Gli1+ progenitor population for enthesis. Please further clarify the innovation of your research, and in depth introduction or discussion is needed to compare and contrast the results between the two research.

Thank you for your suggestions. We discussed the relation and consistency between our dataset and Fang et al. Meanwhile, we integrated our dataset with open source GSE182997 datasets (3 sample) provided by Fang et al. Consequently, we are now better able to discuss the enthesis development from P1 to P58, including 7 timepoints to ideally cover the whole postnatal enthesis growth. (revised in Figure 2 and lines 115-133)

2. In Figure 1, more evidence are needed to prove that Neonatal to postnatal day 7 (P7) is the critical stage for enthesis fibrocartilage cell differentiation, for example, immunofluorescence staining or qPCR for enthesis fibrocartilage cell makers, instead of relying on H and E only.

Thank for your suggestions, we removed the PCA results in revised paper. And we realized the improper statement about P7 as a critical developmental timepoint. We furtherly added toluidine blue staining to observe the extracellular matrix of fibrocartilage. At the same time, we performed immunochemistry to test the protein level of collagen II at enthesis. (revised in Figure 1 and lines 86-104)

3. Line123. Figure 2e showed that the expression of Clec3a and Col2a1 were low in c4," which were ubiquitously expressed in bone-tendon junction cell (c4)" seems to be an inexact expression.

Thank you for your suggestions. In combination with GSE182997 datasets (3 sample) provided by Fang et al., we re-analyzed the clustering outcomes of BTJ chondrocytes and we believed it is inappropriate to subcluster the current BTJ chondrocytes. And we removed the results related to the three enthesis cell clusters in revised version.

4. Line 117, which cell clusters belong to "fibroblast-associated cells"?

Thank you for your suggestions. We are sorry for the misinterpretation of cell annotations. Because we integrated our dataset with open source GSE182997 datasets (3 sample) provided by Fang et al., thus we re-clustered and re-annotated the cell clusters. And the clustering descriptions had been revised in the current version paper. (revised in Figure 2 and lines 115-141)

5. Line 125, it is better to co-staining the scx and Sox9 to validate the existence of BTJ cells. Scx and Sox9 are known markers of BTJ, do you have find new makers for BTJ by scRNA-seq?

Thank you for your suggestions. We co-stained Scx+ and *Sox9*+ and recalculated Scx and *Sox9* positive cells in enthesis. We also checked Scx and *Sox9* expression in our P1 spatial transcriptomic data. (revised in Figure 2e, 2f and lines 134-141)

Meantime, we found expression of Tnn, Clec3a, and Mfge8 was significantly upregulated in enthesis chondrocytes differentiation, suggesting their possibilities as new makers for BTJ chondrocytes, and we checked their expression in spatial transcriptomic data. Furtherly we stained the protein expression of Tnn and Mfge8 to validate their location (revised in Figure 4 and lines 216-225).

In future study, we will label these markers on transgenic mice and investigate their in-vivo function in enthesis development. (revised in lines 412-416).

6. Line 148, "stemness" degree? Are there other evidence, such as stem cell maker expression, to show that "growth plate cells and fibroblasts associated clusters are higher than other cell types". The expression of "stemness" seems exaggerated.

Thank you for your suggestions. We re-analyzed the datasets and we filtered out doublets, dead, and apoptotic cells, blood cells (erythrocytes and progenitors), endothelial cells, immune cells (B cells and T cells), myeloid cells, and growth plate chondrocytes.

The downstream analysis in revised version was focused on the developmental trajectories for tenocytes, chondrocytes, and osteocytes differentiation in enthesis. (revised in Figure 2, 3 and lines 155-225)

7. There is no description of figure 4b in the results.

Thank you for your suggestions. We corrected the mistake in revised manuscription. (revised in Figure 4 and lines 200-203)

8. In figure 5, 2-3 makers identified by scRNA-seq for fibrocartilage formation are suggested to be validated by immunofluorescence stainning or other methods, instead of only proving the Tnn expression in postnatal BTJ growth.

Many thanks for your suggestions. We performed immunofluorescence staining to validate the results from cell trajectory inference (Tnn and Mfge8 expression, figure 4), transcription factor calculation (Mef2a/2c and *Sox9* expression, figure 5), and communication results (Fgf2r and Bmp2r expression, figure 6). And we also performed spatial transcriptome sequencing on enthesis slide at postnatal day 1 to exam the location of these results in enthesis. (revised in Figure 4 and lines 216-225)

9. There are no verification of the signaling network for the enthesis postnatal growth which were revealed by Cellchat. It is suggested to validate the key signaling, such as Bmpr2 signaling.

Thank you for your suggestions, we refined the cellchat input for downstream analysis, including 7 clusters of enthesis-related cells (BTJ chondrocytes, BTJ tendons, tenocytes, osteocytes, enthesoblasts, and mesenchymal progenitors). Then we checked the expression and location of CellChat inferred target genes in spatial transcriptomic data. Meanwhile we performed immunofluorescence staining to check the *Fgfr2* and Bmpr2 expression inferred by cellchat. (revised in Figure 6 and lines 286-303)